# Performance of NeQuick-2 and IRI-Plas 2017 Models during Solar Maximum Years in 2013–2014 over Equatorial and Low Latitude Regions

Kenneth Iluore [1] , Jianyong Lu [1],*, Francisca Okeke [2] and Kesyton Oyamenda Ozegin [3]

1 Institute of Space Weather, School of Atmospheric Physics, Nanjing University of Information Science & Technology, Nanjing 210044, China; kiluore@aul.edu.ng
2 Department of Physics & Astronomy, University of Nigeria, Nsukka 400241, Nigeria; Francisca.okeke@unn.edu.ng
3 Department of Physics, Ambrose Alli University, Ekpoma 310101, Nigeria; ozeginkess@yahoo.com
* Correspondence: jylu@nuist.edu.cn

**Abstract:** This paper carries out a comparative investigation of the Total Electron Content (TEC) values calculated by using the NeQuick-2 and IRI-Plas 2017 models. The investigation was carried out for the solar maximum year of 2013–2014 with data from eight GPS stations within the equatorial and low latitude regions. The results show that both models agree quite well with the observed TEC values obtained from GPS measurements in all the stations, although with some overestimations and underestimations observed during the daytime and nighttime hours. The NeQuick-2 model, in general, performed better in months, seasons, and in most of the stations when the IRI-Plas overestimates the GPS-TEC. However, it is interesting to know that with an increase in solar activity in some seasons, the quality of forecasting IRI-Plas can improve, whereas for the NeQuick-2 model, it decreases, but this is not true for all the seasons and all the stations. Factors causing the discrepancies in the IRI-Plas data model might be caused by the plasmaspheric part included in the IRI, and it is found to be maximum at the MBAR (34%) station, whereas that of the NeQuick-2 data model is found to be maximum at the ADIS (47.7%) station. There is a latitudinal dependence for both models in which the prediction error decreases with the increasing latitudes.

**Keywords:** IRI-Plas 2017 model; NeQuick-2 model: GPS-TEC variation

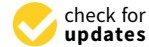



## 1. Introduction

The variation of the Earth's ionosphere is complicated and may behave differently from region to region. The application of the ionospheric communication systems such as satellites, aircraft, and surface transportation systems has increased enormously in recent years. Therefore, the extensive knowledge of the morphology of the ionospheric Total Electron Content (TEC) is of great importance to the scientific community. TEC is defined as the total number of electrons in a vertical column with a cross-sectional area of one square meter centered on the signal path. TEC is measured in TEC Units (1 TECU = $10^6$ electrons per $m^2$) [1]. Vertical Total Electron Content (VTEC) is a very good indicator of the degree of ionization of the Earth's ionosphere and has many practical applications in satellite navigation, time delay, and range error corrections for single frequency GPS satellite signal receivers [2]. The accurate values of TEC are required for making appropriate range corrections, as well as accounting for errors introduced in the range of delays owing to the effects of space weather-related events, such as geomagnetic storms and scintillation due to ionospheric irregularities.

Research studies have been carried out on the variation of GPS-TEC diurnal, monthly, and seasonal variations in low and equatorial latitudes, and empirical models have been developed. In all the empirical models, the International Reference Ionosphere (IRI) model is the most popular ionospheric model and has been recognized as the de facto standard

for specifying ionospheric parameters such as electron temperature, electron density, TEC across the globe [3]. Some findings presented by researchers who compared the model predictions and the GPS-TEC observation show that the model computes the monthly averages of the ionospheric parameters from the altitude ranging from 60 to 2000 km [3–5]. NeQuick model [6–8] is another popular global empirical model which has been used by several researchers to compare with GPS-TEC measurements and has also shown to be a good representation of the ionosphere [5,9–11]. The NeQuick model was proposed by Di Giovanni and Radicella [12] and was subsequently modified by Radicella and Zhang [13] to give vertical TEC from the ground to the orbital height of the satellite GNSS (global navigation satellite systems), consistent with the European Cooperation in the field of scientific and technical research. It describes the electron density beyond the peak of the F2 layer to the GPS satellite altitude by including routines that compute the electron density along any ray-path from the ground to GNSS satellite altitudes of about 20,200 km. The latest version of the model NeQuick-2 was released by the Trieste-Graz Ionospheric profilers [7]. It was developed by implementing further improvements by Radicella and Leitinger [6], a modified bottom side was introduced by Leitinger et al. [14] and a modified top side proposed by Co¨ısson et al. [15]. This model is acceptable because it has improved the estimating of the topside ionosphere, and consequently, versions of the IRI model from 2007 and later have included the topside formulation of the NeQuick, and have adopted it as the most mature of the different proposals. Version 1 of the NeQuick was adopted by International Telecommunication Union Radio Communication Sector (ITU-R) as a procedure for estimating TEC and the NeQuick2 is currently recommended by the ITU Nava and Radicella [16].

Some comparative studies have been made on the performance of the modeled TEC and the observed TEC in different regions using the IRI model and NeQuick model. Chauhan and Singh [17] compared GPS TEC values with that of IRI-2007 at a low latitude station, Agra in India, and showed that GPS-TEC measurements were relatively close to the NeQuick and IRI-01 Corr for local times between 06:00 and 18:00 hr, whereas outside this time sector only the IRI-2001 matched well with GPS-TEC measurements. Kenpankho et al. [18] compared TEC values obtained from IRI-2007 and the GPS receiver at Chumphon Thailand during 2004–2006 at a low latitude station. They showed that GPS-TEC agrees closely with NeQuick and IRI-Corr for local times between 06:00 and 18:00 h, whereas outside this time sector only IRI-2001 matches well with GPS-TEC. Adebiyi et al. [19,20] carried out an assessment of IRI and IRI-Plas models over African equatorial and low latitude region. They found that the diurnal and seasonal structures of modeled TEC follow quite well with the observed TEC in all the stations. Additionally, the IRI-Plas model, in general, performed better in months and seasons when the three options of the IRI model were underestimated. Okoh et al. [21] worked on the assessment of the NeQuick-2 and IRI-Plas 2017 models using global and long-term GNSS measurements. They found both models follow quite well in trends with GNSS measurements. Furthermore, their results showed that NeQuick model performed better than IRI-Plas except in stations located in Antarctica (DAV1), French Southern (KERG), and Ethiopia (ADIS). Zakharenkova et al. [22] compared IRI-2012 and IRI-Plas models with GPS VTEC data from European mid-latitude GPS station Potsdam. They found that the new extension of IRI model (IRI-Plas) overestimates GPS VTEC, especially for low and moderate solar activity, and cannot correctly represent the VTEC variations over European mid-latitudes. Kumar et al. [23] compared the TEC from GPS at Singapore during the year 2010 and 2011 with those derived from the latest IRI-2012 model with three options, IRI-NeQ, IRI-2001, and IRI 2001-Corr, for topside electron density. Their results showed that the IRI-NeQ and IRI01-Corr models are in good agreement with GPS-TEC values at all times, in all seasons, during the year 2010. Tariku [24] carried out analysis on GPS-VTEC variation and compare the results with the latest version (IRI-2012) model and the IRI-NeQuick topside model over equatorial regions of Uganda during a very low and a high solar activity year. His results show that the largest overestimations by the IRI-2012 model was seen during the low solar activity year. This is in contrast to the works of Venkatesh et al. [25], who carried out a comparative study of the

GPS-TEC variations and compared the results with the IRI-2012 and NeQuick-2 modeled TEC over Brazilian equatorial and low latitude sectors during the ascending phase of solar activity from 2010–2013. They found that the performances of the model improved during the low solar activity year compared with that during the high solar activity years [26].

It is interesting to note that some of the results presented by earlier researchers over the different regions are found to be different. However, there could not be found a study that carries out a comparative study of the GPS-TEC measurements, NeQuick-2 and the IRI-Plas modeled data over low and equatorial latitudes regions with different longitudes. Additionally, also, according to the works of Lyon and Thomas [27], the latitudinal position of the EIA is different in different longitudinal sector and shifts with increasing/decreasing the solar activity which is why it is a challenging task for the ionosphere modelers to include the property of EIA in the IRI-model at an adequate level. The validation of NeQuick-2 and IRI-Plas with the GPS-TEC measurement in different longitudes over the equatorial and low latitude region could be beneficial for ionospheric modelers.

Our purpose is to evaluate the performance of the NeQuick-2 model and IRI-Plas 2017 model over eight selected GPS-TEC stations in the equatorial and low latitude region. Our results will show the potential of the European NeQuick and IRI Plas models in predicting TEC values over the region. The TEC calculated by IRI-Plas model involves the electron density integration up to the orbital height of the GNSS satellite (20,200 km). Whereas the IRI model can only compute the ionospheric parameters up to 2000 km, which in this case an underestimation of the true value of TEC will occur because the model does not take into account for electron contents up to the orbital height of the satellite 20,200 km. To overcome this height limitation, we use the IRI-Plas model, which extends to the plasmasphere Gulyaeva and Bilitza [28], can provide a structural model of the ionosphere up to the plasmasphere [29].

The paper is organized as follows. In Section 2, we introduce the data and methods used in this research. Section 3 presents the performance of the NeQuick-2 model and IRI-Plas 2017 model over eight selected GPS-TEC stations within the equatorial and low latitude regions. Section 4 gives a discussion causing these data-model's discrepancies. Additionally, a brief summary and conclusion are given in Section 4.

## 2. Data and Methods

The GPS-TEC observation data in RINEX FORMAT for the years 2013 and 2014 are selected from eight GPS receivers stationed at different locations within the equatorial and low latitude regions (between the Equator and 30º North/South). The TEC data are obtained from the Low latitude Ionospheric Sensor Network (LISN) at (http://lisn.igp.gob.pe/ (accessed on 17 December 2021)); African Geodeotic Reference Frame (AFREF) at (http://www.afrefdata.org/ (accessed on 17 December 2021)), and the Crustal Dynamics Data Information System at (https://cddis.nasa.gov/Data/ (accessed on 17 December 2021)). The geographic and geomagnetic coordinates of the eight stations used for this study are lying at equatorial and low latitudes, but at different longitudes whose locations in the map are shown in Figure 1, whereas the geographic and geomagnetic coordinates are listed in Table 1. Figure 1 shows the map of the world indicating the location of the selected GPS stations within the equatorial and low latitude region. The eight stations are Malinda in Kenya, Mbarara in Uganda, Librevile in Gabon, Cotonou in Benin, Addis Ababa in Ethiopia, Bangalore in India, Dodedo in Guam, and Patumwen in Thailand.

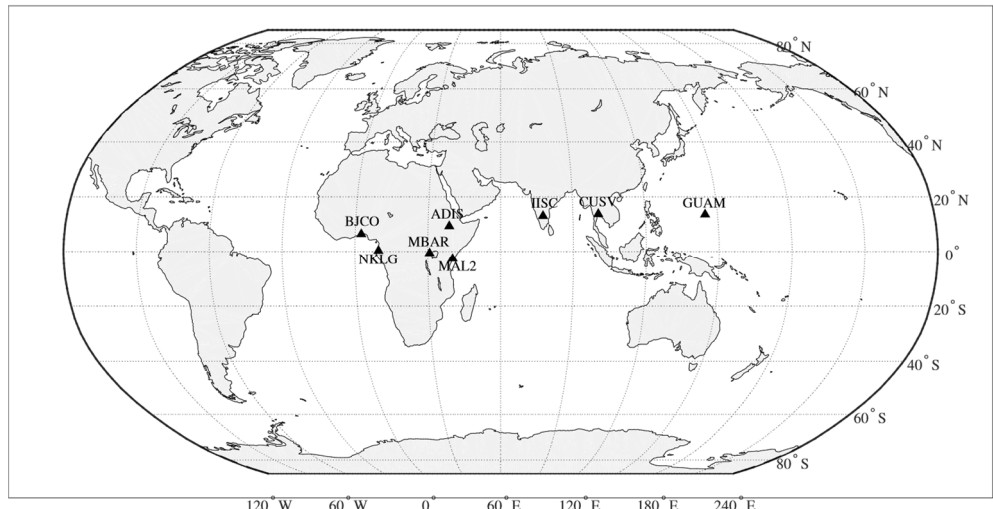

**Figure 1.** Geographic map of the study areas showing the global distribution of the eight GPS stations: MAL2, MBAR, NKLG, BJCO, ADIS, IISC, GUAM, and CUSV.

**Table 1.** List of GPS Stations with Their Geographic and Geomagnetic Coordinates.

| | | Geographic Coordinates | | Geomagnetic Coordinates | |
|---|---|---|---|---|---|
| **Stations/Country** | **Station Code** | **Latitude ($o_N$)** | **Longitude ($o_E$)** | **Latitude ($o_N$)** | **Longitude ($o_E$)** |
| Malinda, Kenya | MAL2 | −2.7 | 40.19 | 6.41 | 111.92 |
| Mbarara, Uganda | MBAR | −0.6 | 30.74 | −10.22 | 102.36 |
| Librevile, Gabon | NKLG | 0.35 | 9.7 | −8.01 | 81.08 |
| Cotonou, Benin | BJCO | 6.4 | 2.5 | −3.07 | 74.59 |
| Addis Ababa, Ethiopia | ADIS | 9.3 | 38.76 | 0.16 | 110.46 |
| Bangalore, India | IISC | 13.02 | 77.57 | 4.49 | 150.93 |
| Dodedo, Guam | GUAM | 13.58 | 144.86 | 5.75 | −143.51 |
| Patumwen, Thailand | CUSV | 13.73 | 100.53 | 4.02 | 173.33 |

The RINEX (Receiver Independent Exchange) format observation files obtained from these stations are processed using the GPS-TEC analysis application software version 2.2 [30] developed by Gopi Seemala of the Institute for Scientific Research, Boston College, USA. This software calculates GPS-TEC based on the principle that two different-frequency of GPS-TEC signals transmitted at the same time from the satellite through the same route will be delayed differently by the ionosphere. The measurements of the two different frequencies of the GPS are analyzed to derive the Slant Tec (STEC) values using the differential delay techniques similar to the works of Seemala and Valladares [31], Oron et al. [32], and Rama Rao et al. [33]. The derived STEC values along the satellite ray path are converted into vertical TEC (VTEC) using a thin shell approximation and considering the ionospheric height of 350 km using the following relation.

$$VTEC = \frac{[TEC_R - (b_R + b_s)]}{S(E)} \tag{1}$$

where S(E) is the single layer mapping function of the ionosphere defined by

$$S(E) = \sec\left\{ \sin{-1} \left[ \frac{(R_e \cos\varepsilon)}{(R_e + h)} \right] \right\} \tag{2}$$

where $R_E$ and $h$ are the Earth's radius (∼6378.1 km) and ionospheric height 350 km, respectively, and $\varepsilon$ is the elevation angle in radians.

The hourly VTECs are averaged and allow us to compute the monthly mean values of the diurnal GPS-TECs. In addition, the NeQuick and IRI-Plas models provide the monthly average values of the ionospheric parameters and compute monthly mean value. The

NeQuick-2 monthly GPS-TEC values were obtained using the Windows executable program created from the FORTRAN source code which was obtained from the Ionospheric Radio Propagation unit of the T/ICT4D laboratory (https://t-ict4d.ictp.it/nequick2/source-code (accessed on 17 December 2021)), whereas the monthly VTEC values of IRI-Plas 2017 were obtained using the Windows executable program which was obtained from the website of the IZMIRAN Institute (http://ftp.izmiran.ru/pub/izmiran/SPIM/ (accessed on 17 December 2021)).

The VTECs are further grouped for seasonal evaluations (i.e., three months of VTEC data are used for each season) for the models, and here the seasons include March equinox (February, March, and April), June Solstice (May, June, and July), September equinox (August, September, and October), and December Solstice (November, December, and January) [34].The mean seasonal TECs are obtained by averaging the diurnal monthly mean values of the months in each division. We compute the Root Mean Square Deviation (RMSD) of the IRI-Plas, NeQuick-2 from the GPS-TEC, and the percentage RMSD (%RMSD) of both models from the GPS-TEC using Equations (3) and (4) [35].

$$RMSD = \sqrt{\frac{\sum_1^n (X_{obs} - X_{model})^2}{n}} \quad (3)$$

where $n$ is the number of data points, $TEC_{obs}$ is the observed TEC Values, and $TEC_{model}$ is the model values of IRI-Plas and NeQuick.

$$\text{Percentage } RMSD = \frac{RMSD}{RMS_{obs}} \times 100, \left[ where\ RMS_{obs} = \sqrt{\frac{\sum_{i=1}^n (obs)^2}{n}} \right] \quad (4)$$

The percentage deviation (%DEV) between IRI-Plas, NeQuick results and the GNSS observed values of the GPS-TEC is also analyzed, according to the following equation,

$$\% D = \frac{[(Model\ prediction - Observe\ GNSSvalue)]}{Observe\ GNSS\ value} \times 100 \quad (5)$$

## 3. Result

Figures 2 and 3 show the models' predictions in comparisons with the GPS-TEC observations, respectively. The red-colored curves represent the GPS-TEC, whereas the blue and the black colored curves represent the predictions of the IRI-Plas and NeQuick models, respectively. These figures are the seasonal evaluations of the GPS-TEC for solar maxima year (2013–2014). The plots layout is divided into four seasons: March equinox (February, March, and April), June Solstice (May, June, and July), September equinox (August, September, and October), and December Solstice (November, December, and January). The empty panels indicate the months where GPS-TEC data are not available. Using the monthly RMSD Equation (3), the monthly errors of the prediction from both models for the solar maximum year are computed and illustrated in Figure 4, which shows the plots of the monthly RMSD against the month. Figures 5 and 6 show the model GPS-TEC relative deviation from the corresponding model values, respectively. These are carried out to investigate the performance of both models according to the seasonal variations and local time. In all, a positive %Dev value indicates that both models overestimate the GPS TEC values while a negative %Dev value indicates that the models underestimate the GPS-TEC values, also at zero or close to zero %Dev depicts more accurate predictions.

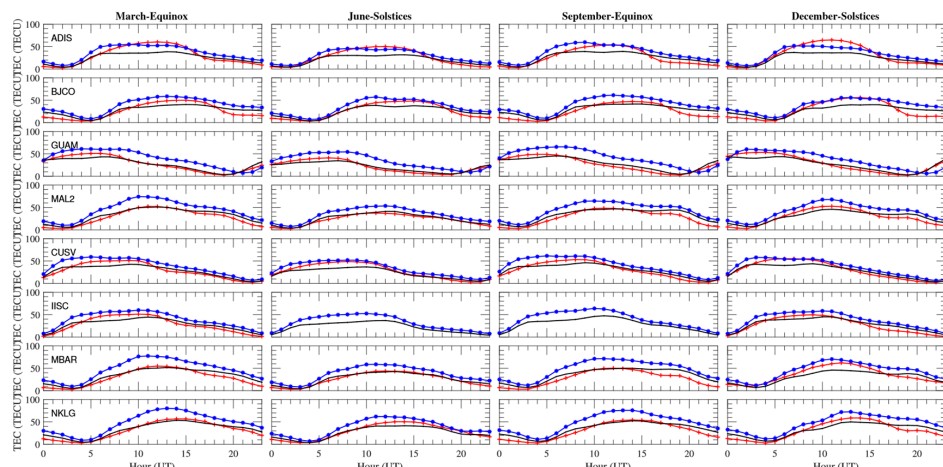

**Figure 2.** Averaged seasonal variation of GPS-TEC with corresponding modeled TEC for the entire stations considered in 2013.

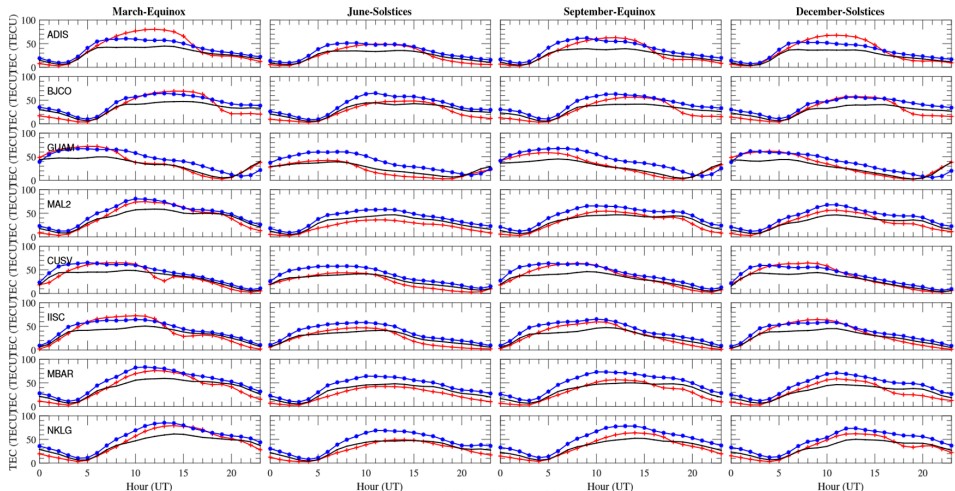

**Figure 3.** Averaged seasonal variation of GPS-TEC with corresponding modeled TEC for the entire stations considered in 2014.

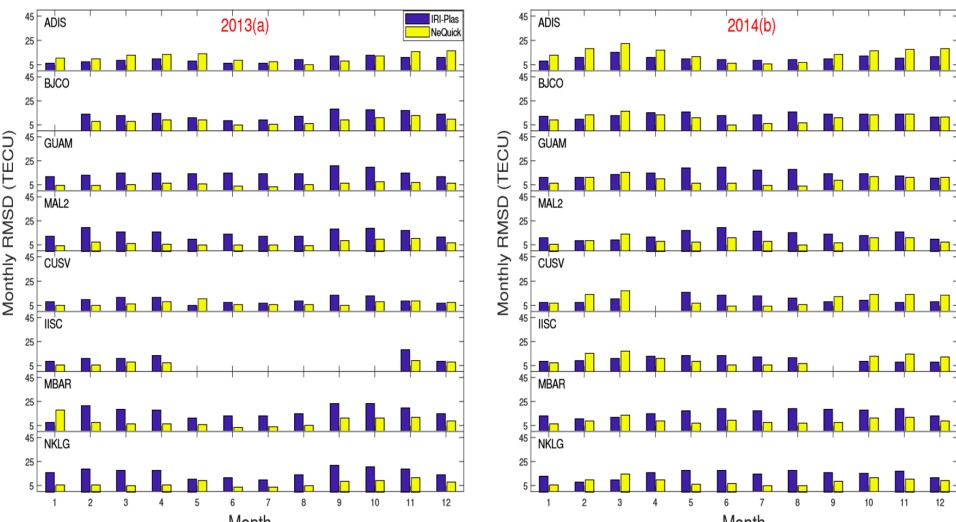

**Figure 4.** Monthly RMSDs for the year 2013 (**a**) and 2014 (**b**) for all the stations.

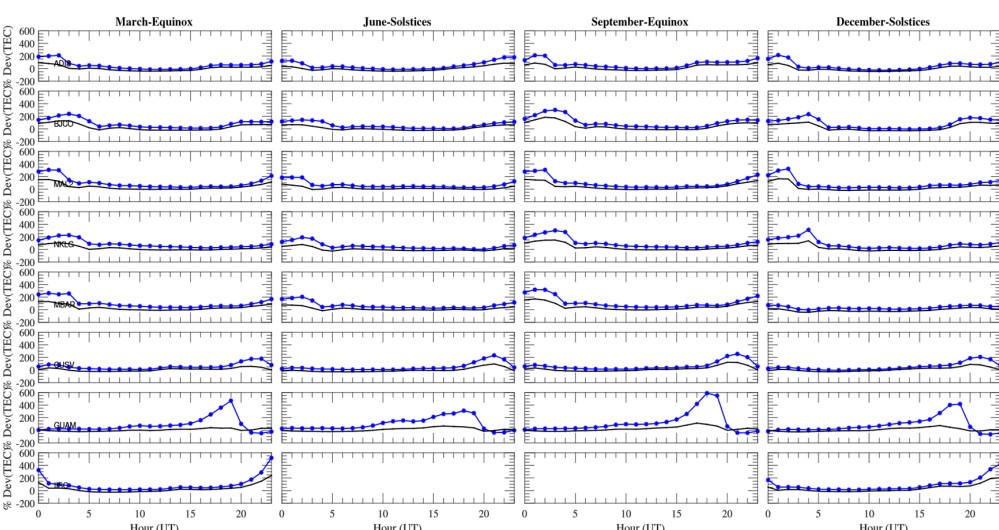

**Figure 5.** Averaged seasonal Percentage Deviation of IRI-PLAS and NeQuick model for the entire stations considered in the year 2013.

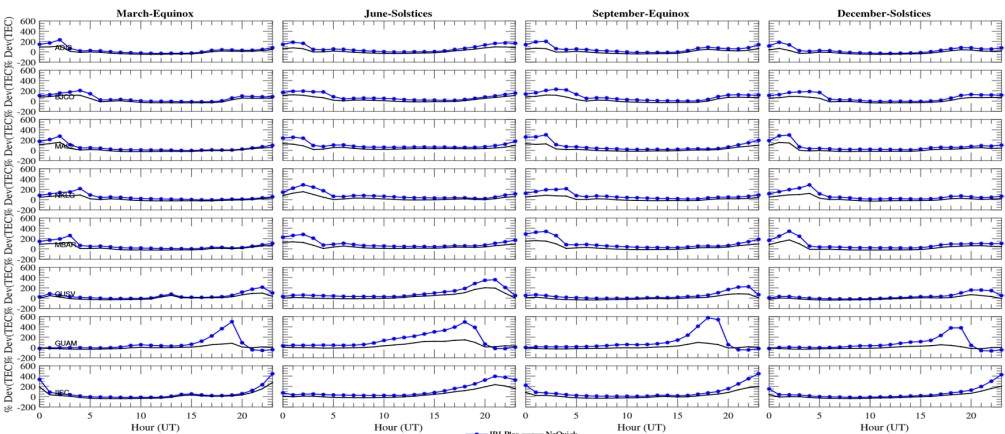

**Figure 6.** Averaged seasonal Percentage Deviation of IRI-PLAS and NeQuick model for the entire stations considered in the year 2014.

### 3.1. Seasonal Comparison of GPS-TEC with NeQuick and IRI-Plas Model at Mbarar in Uganda

In the year 2013, the NeQuick model overestimated the GPS-TEC during the equinoxes with a maximum of approximately 5 TECU in all the time of the day but agreed fairly well during 10.00 UT–14.00 UT. It is also interesting to know that the NeQuick model showed a very good agreement with the GPS-TEC in all the hours of the day during the June solstice but underestimated between 0.00 UT to 17.00 UT with a maximum of approximately −18 TECU (Figure 5) at around 12.00 UT and overestimated from 19.00 UT to 24.00 UT during December solstices. Thus, in 2014, the NeQuick model showed a good correlation during the buildup period (i.e., 4.00 UT–8.00 UT) in all the seasons but underestimated the GPS-TEC during the daytime hours between 9.00 UT to 19.00 UT and overestimated at nighttime. In addition, the model also overestimated the GPS-TEC in all the hours of the day with a maximum of approximately 3 TECU at around 12.00 UT during the June solstice. The IRI-Plas model overestimates the GPS-TEC during all the times and seasons with a maximum of approximately 20 TECU.

### 3.2. Seasonal Comparison of GPS-TEC with NeQuick and IRI-Plas Model at Malinda in Kenya

The NeQuick model overestimated the GPS-TEC values during the whole day in all the seasons except in the September equinox when the model underestimated during

the daytime hours (maximum approximately at −10 TECU at around 10.00 UT) and overestimated at nighttime (17.00 UT–24.00 UT). It is also interesting to know that the model also showed a good agreement with GPS-TEC during the period of (9.00 UT–16.00 U T) in the September equinox, (9.00 UT–11.00 UT and 17.00 UT–24.00 UT) in the June solstice, and (13.00 UT–15.00 UT) in the September equinox. In 2014, the model agreed fairly well with the GPS measurement during the buildup period (4.00 UT–7.00 UT) in all the seasons and underestimated GPS-TEC during the noontime hours (9.00 UT–16.00 UT). However, overestimation was observed in the June solstice (5 TECU) in all the daytime hours. For both years, the IRI-Plas model overestimated the GPS-TEC measurement in all the times and seasons with a maximum approximately at 20 TECU during the March equinox of 2013 and a minimum overestimation approximately at 10 TECU during the March equinox of 2014 but showed a good agreement for the March equinox at 14.00 UT–17.00 UT during the year 2014.

### 3.3. Seasonal Comparison of GPS-TEC with NeQuick and IRI-Plas Model at Librevile in Gabon

Considering the results of NKLG station (i.e., Gabon) in 2013, the NeQuick underestimated and overestimated the GPS measurement during the daytime hours (10.00 UT–18.00 UT) and nighttime (19.00–24.00 UT) in all the seasons except in the June solstice, when the model agreed fairly well with the GPS-TEC during the buildup period (4.00 UT–10.00 UT) and underestimated in the remaining hours of the day. In 2014, the NeQuick model showed a good agreement with the GPS-TEC values during the whole day in the June solstice, except during 11.00 UT–19.00 UT in the remaining seasons during the daytime hours when underestimation was observed and it was at maximum −5 TECU (Figure 5). For both years, the IRI-Plas overestimated GPS-TEC values during all the times and seasons with a maximum of approximately 20 TECU during the March equinox in the year 2013 and a minimum overestimation approximately at 15 TECU during the March equinox in 2014 (Figure 6).

### 3.4. Seasonal Comparison of GPS-TEC with NeQuick and IRI-Plas Model at Cotonou in Benin

During the year 2013 and 2014, the NeQuick model showed a very good agreement during the buildup period (0.0 UT–6.00 UT) and underestimated the GPS-TEC during all the seasons. For the March equinox, the maximum was approximately −6 TECU at around 15.00 UT. The June solstice maximum was approximately −4 TECU, the September equinox maximum was approximately −3 TECU, and the December solstice maximum was approximately −6 TECU during the daytime hours. The IRI-Plas model overestimated the GPS-TEC during all time and seasons except during 11.00 UT–17.00 UT for the December solstice in both years where the model agreed fairly well with the GPS-TEC and underestimated during the March equinox at 14.00 UT–18.00 UT in 2014.

### 3.5. Seasonal Comparison of GPS-TEC with NeQuick and IRI-Plas Model at Addis Ababa in Ethiopia

In the year 2013, the NeQuick model agreed fairly well with the GPS-TEC during the build-up period, i.e., 3.00 UT–7.00 UT. Underestimation and overestimation was observed during the noontime hours (8.00 UT–18.00 UT), with a maximum approximation of −5 TECU. For the year 2014, the NeQuick model underestimated the GPS-TEC during the noontime hours in all the seasons, with maximum approximation 5 and −10 TECU. In both years, the IRI-Plas underestimated the GPS-TEC in all the seasons during the day time hours but agreed fairly well with the GPS-TEC in the September solstices from 11.00 UT–14.00 UT and the June solstice from 11.00 UT–15.00 UT.

### 3.6. Seasonal Comparison of GPS-TEC with NeQuick and IRI-Plas Model at Bangalore in India

During the year 2013, the NeQuick model showed a very good agreement during the buildup period (0.0 UT–4.00 UT) but underestimates the GPS-TEC during the noon time hours (5.00 UT–13.00 UT) with a maximum of approximately −5 TECU at around 12.00 UT (Figure 2) and overestimates at nighttime in the March equinox and the December solstice.

There are no GPS-TEC data in the June solstice and the September equinox. In 2014, the NeQuick model showed an agreement during the buildup period between 4.00 UT and 8.00 UT but underestimated the GPS-TEC in all the seasons except from 6.00 UT to 24.00 UT when an overestimation was observed in the June solstice. The overestimation reached the maximum at approximately 4 TECU at around 12.00 UT. In both years, the IRI-Plas overestimated the GPS-TEC at maximum in all the seasons at approximately 8 TECU at around 12.00 UT, except from 6.00 UT to 13.00 UT in the March equinox and the December solstice when an underestimation was observed.

### 3.7. Seasonal Comparison of GPS-TEC with NeQuick and IRI-Plas Model at Dodedo in Guam

During the year 2013 and 2014, the NeQuick model underestimated between 0.00 UT to 9.00 UT and the maximum was approximately −3 TECU at around 8.00 UT (Figure 5) and overestimated between 14.00 UT and 24.00 UT in all the seasons except in the March equinox from 10.00 UT to 13.00 UT when the model correlated well with the GPS TEC. The IRI-Plas overestimated the GPS-TEC in all the times and seasons of both years, except in the March equinox 2014 from 0.00 UT to 5.00 UT and the December solstice from 2.00 UT to 8.00 UT when the model correlated well with the GPS-TEC.

### 3.8. Seasonal Comparison of GPS-TEC with NeQuick and IRI-Plas Model at Patumwen in Thailand

In the year 2013, the NeQuick model showed an underestimation during 0.00 UT to 13.00 UT with a maximum of approximately −5 TECU at around 13.00 UT in all the seasons and correlates well during the remaining hours from 14.00 UT to 24.00 UT. For the year 2014, the NeQuick model underestimated during the period between 3.00 UT and 12.00 UT in all the seasons and correlated well at nighttime, but in the June solstice, the model showed a good correlation during 0.00 UT–19.00 UT and underestimated the GPS-TEC in the remaining hours of the day. In 2013, the IRI-Plas model overestimated the GPS TEC during all the times and seasons with a maximum of approximately 12 TECU except the period between 5.00 UT and 11.00 UT when the model showed a good agreement with the GPS-TEC in December solstice. In the year 2014, the IRI-Plas overestimated in all the seasons except in the September equinox when the model correlated well with the GPS-TEC between 7.00 UT and 13.00 UT. It is also interesting to know that IRI-Plas underestimated between 6.00 UT and 11.00 UT for the December Solstice.

## 4. Discussion

The Performance of the NeQuick-2 and IRI-Plas 2017 models to predict the GPS-TEC over eight selected stations within the equatorial and low latitude region for the year 2013 and 2014 has been investigated. The RMSD, seasonal, and annual values of %RMSD during 2013 to 2014 are listed in Tables 2–4. It is found that the %RMSD with the IRI-Plas in stations within the equatorial region (MBAR, NKLG, and MAL2) is significantly larger than those with the NeQuick- 2 model while in stations outside the equatorial region, the %RMSD with the NeQuick-2 is larger than those with the IRI-Plas. The annual mean %RMSD in the IRI-Plas model in stations within the equatorial region is at maximum at MBAR station (34%) during the year 2013 and decreases continuously from 34% to 32% as it moves towards a high solar activity year 2014, whereas in the stations outside the equation region, the annual mean %RMSD of the NeQuick-2 model decreases with increase in latitudes, except in ADIS when the RMSD value is 47.7% in the year 2014. In all, there are some similarities as well as some divergence between the modeled and the observed GPS-TEC values. The divergence between the observed GPS-TEC and the TEC predicted by both models involves the integration of the vertical electron density profile. This has been partially attributed to the inaccurate prediction of the shape of the N(h) Profile, erroneous prediction of the plasmaspheric contribution to the GPS-TEC, lack of proper plasmaspheric model in topside representation, and the latitudinal and Longitudinal variations as reported by other authors [5,19,20,36–38]. We observed from the seasonal plot of stations outside the equatorial region that the NeQuick model generally underestimates the observed GPS-

TEC during the daytime hours. This corresponds to the seasons when the IRI-Plas model performed better than the NeQuick model. This may suggest that the underestimations of the observed GPS-TEC by the NeQuick model in all the seasons may be due to the lack of proper plasmaspheric model in the topside representation [5]. It is also interesting to know that the performance of NeQuick model decreases with increase in latitudes, also the RMSD values gets higher as it moves towards a high solar activity year 2014 with ADIS station having the highest RMSD of 47.77%. In addition, we notice in all the seasons that the IRI-Plas model predictions are higher in (MBAR, NKLG, and MAL2), and this corresponds to the stations where the NeQuick predictions are better. In the works of Ezquer et al. [5] in comparing the IRI-Plas with the NeQuick model, they suggested that one of the reasons of overestimation of the IRI-Plas model could be due to the erroneous prediction of the plasmaspheric contribution to the vertical total electron content. The results from our study could be helpful to strategically improve the predictions of the IRI-Plas model.

**Table 2.** Root Mean Square Deviations (RMSD) of the IRI-Plas and NeQuick-2 TEC values from the GPS-TEC values for the four seasons at different stations during 2013 and 2014.

| Year | Station | Seasons | IRI-Plas | NeQuick-2 | Year | Station | Seasons | IRI-Plas | NeQuick-2 |
|------|---------|---------|----------|-----------|------|---------|---------|----------|-----------|
| 2013 | MAL2 | MAREQUI | **17.07** [a] | 19.12 | 2014 | MAL2 | MAREQUI | 9.56 | 9.97 |
|  |  | JUNSOLS | 11.91 | **4.79** |  |  | JUNSOLS | 17.4 | **8.7** |
|  |  | SEPEQUI | 16.43 | **7.51** |  |  | SEPEQUI | 13.9 | **7.51** |
|  |  | DECSOLS | 13.48 | **7.04** |  |  | DECSOLS | 12.07 | **7.71** |
|  | MBAR | MAREQUI | 19.12 | **6.29** |  | MBAR | MAREQUI | 12.2 | **10.37** |
|  |  | JUNSOLS | 12.29 | **4.21** |  |  | JUNSOLS | 17.99 | **7.85** |
|  |  | SEPEQUI | 29.59 | **9.24** |  |  | SEPEQUI | 18.49 | **8.59** |
|  |  | DECSOLS | 17.14 | 10.26 |  |  | DECSOLS | 15.02 | **9.07** |
|  | NKLG | MAREQUI | 17.92 | **5.46** |  | NKLG | MAREQUI | **11.06** | 11.33 |
|  |  | JUNSOLS | 10.44 | **5.23** |  |  | JUNSOLS | 16.59 | **5.86** |
|  |  | SEPEQUI | 18.97 | **7.47** |  |  | SEPEQUI | 16.21 | **8.37** |
|  |  | DECSOLS | 16.24 | **8.23** |  |  | DECSOLS | 13.84 | **8.34** |
|  | BJCO | MAREQUI | 13.36 | **8.27** |  | BJCO | MAREQUI | **12.67** | 14.22 |
|  |  | JUNSOLS | 9.33 | **6.55** |  |  | JUNSOLS | 13.94 | **7.11** |
|  |  | SEPEQUI | 16.05 | **8.45** |  |  | SEPEQUI | 14.4 | **10.38** |
|  |  | DECSOLS | 15.32 | **10.98** |  |  | DECSOLS | 12.68 | **11.4** |
|  | ADIS | MAREQUI | **8.68** | 11.83 |  | ADIS | MAREQUI | **12.44** | 19.18 |
|  |  | JUNSOLS | **6.94** | 10.06 |  |  | JUNSOLS | 9.16 | **7.93** |
|  |  | SEPEQUI | 11.42 | **8.55** |  |  | SEPEQUI | **10.4** | 12.19 |
|  |  | DECSOLS | **9.4** | 14.22 |  |  | DECSOLS | **10.21** | 16.04 |
|  | IISC | MAREQUI | nan | nan |  | IISC | MAREQUI | nan | **10.85** |
|  |  | JUNSOLS | nan | nan |  |  | JUNSOLS | nan | 12.67 |
|  |  | SEPEQUI | nan | nan |  |  | SEPEQUI | nan | nan |
|  |  | DECSOLS | 8.58 | **5.57** |  |  | DECSOLS | **8.11** | 13.2 |
|  | GUAM | MAREQUI | 14.15 | **5.08** |  | GUAM | MAREQUI | **13.26** | 11.97 |
|  |  | JUNSOLS | 14.46 | **4.3** |  |  | JUNSOLS | 18.69 | **5.77** |
|  |  | SEPEQUI | 18.21 | **6.4** |  |  | SEPEQUI | 15.39 | **8.22** |
|  |  | DECSOLS | 12.76 | **2.98** |  |  | DECSOLS | 11.29 | **9.49** |
|  | CUSV | MAREQUI | 10.98 | **6.41** |  | CUSV | MAREQUI | **8.89** | 15.65 |
|  |  | JUNSOLS | **6.48** | 7.37 |  |  | JUNSOLS | 13.99 | **5.36** |
|  |  | SEPEQUI | 11.72 | **6.15** |  |  | SEPEQUI | **9.45** | 10.66 |
|  |  | DECSOLS | 7.94 | **7.03** |  |  | DECSOLS | **7.59** | 11.5 |

[a] Note: The numbers in bold indicate the value of the model with the lowest prediction error.

**Table 3.** % Root Mean Square Deviations (%RMSD) of the IRI-Plas and NeQuick-2 TEC Values from the GPS-TEC Values for the four Seasons at different stations during 2013 and 2014.

| Year | Station | Seasons | IRI-Plas | NeQuick-2 | Year | Station | Seasons | IRI-Plas | NeQuick-2 |
|---|---|---|---|---|---|---|---|---|---|
| 2013 | MAL2 | MAREQUI | 35.23 | 17.74 | 2014 | MAL2 | MAREQUI | 17.69 | 23.69 |
| | | JUNSOLS | 33.18 | 17.51 | | | JUNSOLS | 42.57 | 27.34 |
| | | SEPEQUI | 34.75 | 20.37 | | | SEPEQUI | 29.48 | 20.55 |
| | | DECSOLS | 30.09 | 21.39 | | | DECSOLS | 26.64 | 23.68 |
| | MBAR | MAREQUI | 37.06 | 19.22 | | MBAR | MAREQUI | 21.36 | 24.03 |
| | | JUNSOLS | 30.93 | 14.19 | | | JUNSOLS | 40.13 | 23.18 |
| | | SEPEQUI | 39.69 | 23.79 | | | SEPEQUI | 35.44 | 22.47 |
| | | DECSOLS | 28.57 | 41.37 | | | DECSOLS | 31.17 | 27.07 |
| | NKLG | MAREQUI | 34.29 | 15.16 | | NKLG | MAREQUI | 18.94 | 26.4 |
| | | JUNSOLS | 25.45 | 17.52 | | | JUNSOLS | 35.29 | 17.03 |
| | | SEPEQUI | 36.23 | 19.81 | | | SEPEQUI | 30.49 | 22.11 |
| | | DECSOLS | 33.06 | 22.84 | | | DECSOLS | 27.43 | 23.42 |
| | BJCO | MAREQUI | 36.95 | 27.89 | | BJCO | MAREQUI | 27.39 | 40.67 |
| | | JUNSOLS | 24.67 | 23.44 | | | JUNSOLS | 32.68 | 21.71 |
| | | SEPEQUI | 37.11 | 26.72 | | | SEPEQUI | 32.74 | 32.88 |
| | | DECSOLS | 35.78 | 34.41 | | | DECSOLS | 31.03 | 38.23 |
| | ADIS | MAREQUI | 22.13 | 42.63 | | ADIS | MAREQUI | 29.07 | 59.04 |
| | | JUNSOLS | 21.23 | 42.94 | | | JUNSOLS | 25.46 | 29.62 |
| | | SEPEQUI | 28.46 | 29.07 | | | SEPEQUI | 25.32 | 41.56 |
| | | DECSOLS | 26.11 | 54.3 | | | DECSOLS | 23.32 | 60.87 |
| | IISC | MAREQUI | 27.37 | 22.38 | | IISC | MAREQUI | 22.99 | 39.81 |
| | | JUNSOLS | nan | nan | | | JUNSOLS | 31.62 | 22.94 |
| | | SEPEQUI | nan | nan | | | SEPEQUI | 22.63 | 29.41 |
| | | DECSOLS | 27.76 | 23.72 | | | DECSOLS | 20.17 | 35.46 |
| | GUAM | MAREQUI | 32.64 | 17.74 | | GUAM | MAREQUI | 27.44 | 33.11 |
| | | JUNSOLS | 39.31 | 18.67 | | | JUNSOLS | 44.89 | 21.35 |
| | | SEPEQUI | 39.57 | 20.26 | | | SEPEQUI | 33.09 | 25.25 |
| | | DECSOLS | 31.29 | 19.6 | | | DECSOLS | 27.27 | 30.62 |
| | CUSV | MAREQUI | 25.85 | 20.7 | | CUSV | MAREQUI | 19.99 | 44.58 |
| | | JUNSOLS | 17.98 | 28.44 | | | JUNSOLS | 33.89 | 18.07 |
| | | SEPEQUI | 26.22 | 18.85 | | | SEPEQUI | 20.93 | 32.07 |
| | | DECSOLS | 19.89 | 22.3 | | | DECSOLS | 18.57 | 36.22 |

**Table 4.** Annual Percentage RMSD.

| Year | Station | IRI-Plas | NeQuick-2 | Year | Station | IRI-Plas | NeQuick-2 |
|---|---|---|---|---|---|---|---|
| 2013 | MAL2 | 33.31 | 19.25 | 2014 | MAL2 | 29.09 | 37.43 |
| | MBAR | 34.06 | 24.64 | | MBAR | 32.03 | 24.19 |
| | NKLG | 32.26 | 18.83 | | NKLG | 28.04 | 22.24 |
| | BJCO | 33.63 | 28.12 | | BJCO | 30.96 | 33.37 |
| | ADIS | 24.5 | 42.24 | | ADIS | 25.8 | 47.77 |
| | IISC | 27.56 | 23.05 | | IISC | 24.35 | 31.91 |
| | GUAM | 35.7 | 19.07 | | GUAM | 33.17 | 27.58 |
| | CUSV | 22.49 | 22.57 | | CUSV | 23.35 | 32.74 |

In our current study, the poor performance of the IRI-Plas model within the equatorial region may also be attributed to the erroneous predictions of the plasmaspheric contribution to the GPS-TEC in the region. This plasmaspheric contribution is found to decrease with an increase in the latitudes. This contribution varies from maximum of about 34% in MBAR, 32% in NKLG, and 33% in MAL2 in the year 2013, to the minimum at about 32% in MBAR, 28% in NKLG, and 29% MAL2 in 2014 during all the times of day. These findings are in agreement with earlier studies: Srinivas et al. [39] shown that the plasmaspheric contribution to TEC in this region is up to 30% during the daytime, Klimenko et al. [38] reported a nighttime maximum contribution of 85% and a daytime contribution of 40%

at the equator during the winter in the year 2009 (a period of extreme solar minimum condition). Balan et al. [37] reported a contribution of about 12% during the daytime and 60% during the nighttime over Japan, Lunt et al. [36] and Klimenko et al. [38] both proposed that the plasmaspheric contribution to TEC is decreasing with an increase in latitude and altitude.

Additionally, several other works, Yizengaw et al. [40] and Klimenko et al. [38] achieved similar conclusions. On average, the NeQuick model is observed to be consistent and performed better than the IRI-Plas in all the selected eight GPS stations considered, within the equatorial and low latitude region. In the case of the IRI-Plas model, the predictions are observed to get better during the periods of increased solar activity 2014 when the GPS-TEC are relatively enhanced and performs better in some stations. The model's percentage error drops from higher values during the year 2013 to lower values during higher solar activity year 2014, as shown in Tables 3 and 4. This result is in agreement with the earlier works of Ezquer et al. [5].

The best-case scenario for the NeQuick model is seen in the year 2013 at an equatorial station of NKLG, where the monthly RMSDs is less than 5 TECU for all the months (except for the month of May, Sept, Oct, and Nov) where the values are higher than 5 TECU. The best-case scenario for the IRI-Plas model is seen in the year 2013 at the ADIS and CUSV station, where in all the months in both stations, the RMSD is observed to be less than 13 TECU. The worst-case scenario for the NeQuick model is in the year 2014 in ADIS station when the monthly RMSD reaches 24 TECU for the month of March. For the IRI-Plas, the worst-case scenario is in the year 2013 at Guam station, where the RMDS reaches 25 TECU in the month of September.

## 5. Summary and Conclusions

We have investigated the performance of NeQuick-2 and IRI-Plas 2017 model in predicting the GPS-TEC up to the orbital height of the GNSS satellite (20,200 km) over eight selected GPS stations located in the equatorial and low latitude region during the years 2013–2014. The results from this study show that the TEC predicted by both models agrees quite well with the observed GPS-TEC measurements, although with some upward and downward offset observed during both the daytime and nighttime. The NeQuick-2 model performed better than the IRI-Plas model in most of the months, seasons, and stations when the IRI-Plas overestimates the GPS-TEC. In all, our results show that with an increase in solar activity in some seasons, the quality of forecasting IRI-Plas can improve, while for the NeQuick-2 model, it decreases, but this is not true for all the seasons and not for all of the stations. The GPS-TEC deviations are greater during the equinox than the solstice seasons. It is highly interesting to know that the discrepancy of the models depends on the local time, latitude, and strength of the solar activity. This throws a deep light to the understanding of TEC calculations and employing NeQuick-2 or/and IRI-Plas 2017 model(s). In addition, the analyses of the prediction errors and the annual %RMSD so far established can help in getting results with predictions error that are more accurate and that will yield a more robust result. It is therefore recommended that more work be carried out to really ascertain the validity of these models.

**Author Contributions:** Conceptualization, K.I., J.L., F.O. and K.O.O.; methodology, K.I.; software, K.I.; validation, K.I.; formal analysis, K.I., J.L., F.O. and K.O.O.; investigation, K.I., J.L., F.O. and K.O.O.; resources, K.I.; data curation, K.I.; writing—original draft preparation, K.I.; writing—review and editing, J.L., F.O. and K.O.O.; visualization, K.I.; supervision, J.L., F.O. and K.O.O.; project administration, J.L.; funding acquisition, J.L. All authors have read and agreed to the published version of the manuscript.

**Funding:** This work was funded by National Natural Science Foundation of China (grants 42030203, 41974190).

**Data Availability Statement:** The TEC data are obtained from the Low latitude Ionospheric Sensor Network (LISN) at http://www.jro.igp.gob.pe/lisn/ (accessed on 17 December 2021); African

Geodeotic Reference Frame (AFREF) at http://www.afrefdata.org/ (accessed on 17 December 2021), and the Crustal Dynamics Data Information System at https://cddis.nasa.gov/Data/ (accessed on 17 December 2021).

**Acknowledgments:** The Authors are grateful to the Low Latitude Ionospheric Sensor Network (LISN), the African Geodetic Reference Frame (AFREF) and CDDIS: NASA's Archive of Space Geodesy Data for GPS Data and Product. We also appreciate the NeQuick working group for providing the FORTRAN source code NeQuick-2 using the Windows executable program created which obtained from the ionosphere Radio Propagation unit of the T/ICT4D laboratory of the Abdus Salam International Center for Theoretical Physics, Trieste. Italy. We appreciate the IZMIRAN Institute for providing the FORTRAN source code for IRI-Plas 2017, which is standard plasmasphere Ionospheric Model (IRI-Plas-SPIM) at (http://ftp.izmiran.ru/pub/izmiran/SPIM/ (accessed on 17 December 2021)).

**Conflicts of Interest:** The authors declare no conflict of interest. The funders had no role in the design of the study; in the collection, analyses, or interpretation of data; in the writing of the manuscript, or in the decision to publish the results.

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
