# Peer review of "Performance of NeQuick-2 and IRI-Plas 2017 Models during Solar Maximum Years in 2013–2014 over Equatorial and Low Latitude Regions"

_universe, doi:10.3390/universe8020125_

Round 1

Reviewer 1 Report

Review of the article Kenneth Iluore et al. «Performance of NeQuick-2 and IRI-Plas 2017 Models during  Solar Maximum years in 2013-2014 over Equatorial and Low latitude Regions».

The manuscript is clear, relevant for the field and presented in a well-structured manner.  The manuscript  is scientifically sound and is the experimental design appropriate to test the hypothesis.

The main disadvantage of the article  - the conclusions are not consistent with the evidence and arguments presented.

Аuthors write that «In general, the NeQuick model predictions values are better in equatorial stations (NKLG and MBAR) while the IRI-Plas  model predictions are better in stations outside the equatorial regions except in (GUAM) station.» lines 408-410.

At the same time, Table 4 (line 645) «Аnnual Percentage RMSD» shows that in 2013, out of 8 stations, 7 have less discrepancy for the model NeQuick-2. Even in 2014, where the coincidence of the model  NeQuick-2  prediction and measurements becomes worse, for 3 out of 8 stations RMSD  is still less for the model NeQuick-2 .

If we consider the data separately by seasons, then not only GUAM does not correspond to the statement made that the IRI-Plas  model works better:  IISC - for 3 out of 6 RMCD (two seasons in 2013 and 4 in 2014), for CUSV - in 3 seasons out of  8 considered , BJCO – 5  out of 8 RMCD for NeQuick-2   less than RMSD for IRI-Plas  .

So based on the data in Tables 2, 3, 4, as well as Figures 2-6, the conclusion that the IRI-Plas  model performs better than NeQuick-2  outside the equatorial region is incorrect.

The data presented in the article show that with an increase in solar activity in some seasons, the quality of forecasting IRI-Plas  can improve, while for NeQuick-2, on the contrary, it decreases, but this is not true for all seasons and not for all stations. The authors, if they want to reliably prove the improvement in the performance of the IRI-Plas  model with an increase in solar activity compared to  NeQuick-2 one, should calculate for each of the considered seasons the number of flares above a certain class of flare and SEP events for each of the considered seasons.

The figures and tables are appropriate and properly show the data, except Fig.7. Figures 1 – 6 are  easy to interpret and understand, but for Figure 7 it is necessary to describe the methodology for its creation, since fig. 2, 3, and 5 in no way show that the NeQuick-2 model underestimates forecasts by 30 percent (7a), and the deviations of the IRI-Plas model from the measured values are close to zero, as shown in Fig. 7b. The other data interpreted appropriately and consistently throughout the manuscript.

Authors should clearly indicate which stations they refer to equatorial and which to low-latitude regions.

Table 2 is not mentioned in the text of the article.

Line 637 says « The numbers in Bold indicates the value of the model with the lowest prediction error». But in the table above there is no bold font, there is only a change in font size and type. 

The conclusions need corrected, and the method for obtaining figure 7 need explained, or this figure is excluded, because without explanation it looks contradictory to the rest of the figures.

Reviewer 2 Report

This paper evaluates the performance of NeQuick-2 and IRI-Plas 2017 model in predicting the GPS-TEC up to the orbital height of the satellite (20,200 km) over eight stations located in the equatorial and low latitude region during the year 2013-2014. The results show that both models agree fairly well with observed TEC values obtained from GPS measurements. The NeQuick model performs better in equatorial stations (NKLG and MBAR) while the IRI-Plas model predictions are better in stations outside the equatorial regions except in (GUAM). The reasons leading to the different behaviors of these two models are also discussed. Although these results are not very innovative, they are still interesting. However, there are a lot of writing flaws in the present version that need to be improved before further consideration of being accept. Here are some examples, but definitely not all:

  1. Line 16, “agree fairly well with observed TEC values” --> “agree fairly well with the observed TEC values”.
  2. Line 19, “except in (GUAM)”should be “except in GUAM”?
  3. Line 19, “The reason causing these data-model discrepancies”--> “The reason causing the discrepancies of these data-models”;
  4. Line 22, “at equatorial station”--> “at the equatorial station”;
  5. Line 23, “at station outside”--> “at the station outside”;
  6. Line 42-43, “space weather-related 50 events”, delete the “50”?
  7. Line 46, “In all the empirical Models”--> “In all the empirical models”;
  8. Line 58-59, “from ground 66 to the”, what does 66 means here?
  9. Line 60, “234 regional electron content model”, here “model”should be “models” as their number is 238, right?
  10. Line 64-65, “It was as developed”, delete “as”;
  11. Line 76-77, at a low latitude station, e. Agra in India,and showed...
  12. Line 81, “it showed that”--> “They showed that “;
  13. Line 85, “They observed that the diurnal and seasonal structures of modeled TEC” --> “They found that the diurnal and seasonal structures of the modeled TEC”;
  14. Line 103, Our results will show...;
  15. Line 108, because the model does not take into account the electron contents;
  16. Line 114, “NeQuick-2 model 123”, delete “123”;
  17. Line 115, in the northern and southern;
  18. Line 116, these data-model’s discrepancies;
  19. Line 119, “years 2013 and 2014 128”, delete the number “128”;
  20. Line 121, “and low latitude region”---> “and low latitude regions”;
  21. Line 126, “whose locations in the map as shown in Fig.1 1 while...”-->”whose locations in the map are shown in Fig. 1, while...”;
  22. Line 143, “transmitted”should be “transmitting”?
  23. Line 155-156, “RE=Radius of the Earth”--> “RE is the radius of the Earth”;
  24. Line 198, “represent the variations”--> “represent the predictions”;
  25. Line 210, “the models underestimates”--> “the models underestimate ”;
  26. All times in this paper are written as “10.00 UT - 14.00 UT”(Line 219). They should be written as “10:00UT -14:00 UT” as usual;
  27. “where”in Line 234 should be “when”;
  28. There are both “nighttime”(Line 226) and “night time” (Line 236), they should be unified;
  29. Line 249, “Considering the results of NKLG station, Gabon. In 2013, the NeQuick...”---> “Considering the results of NKLG station (i.e. Gabon) in 2013, the NeQuick...”;
  30. Line 256, “at approximately at -5 TECU”should be “at approximately at -5 TECU”?
  31. Line 304, “from 10.00 UT - 13.00 UT”should be “from 10:00 UT to 13:00 UT” or “during 10:00 UT - 13:00 UT”;
  32. Line 328-329, “It is observed that”-->”It is found that”;
  33. Line 360, “the predictions of IRI-Plas model”--->”the predictions of the IRI-Plas model”;
  34. Line 380, “In the case of IRI-Plas mode”-->”In the case of the IRI-Plas mode”;
  35. Line 381, “where”should be “when”;
  36. Line 385, “than NeQuick model in stations”-->”than the NeQuick model in those stations”;
  37. Line 386, “periods of increased in solar activity, year 2014”-->”periods of the rising time (year 2014) of solar activity”, many other similar errors;
  38. Line 391-392, “in prediction the TEC”-->”in predicting the TEC”.

Reviewer 3 Report

Major comments

  • Both the motivation and the research subject of this paper are very unclear. I do not understand why the authors chose the very IRI-Plas and NeQuick-2 models for their research? What is the novelty of this research? Are the authors intending to find any specific features of the TEC behavior or, vice versa, any features of the ionospheric models? Probably, the authors offer modified method of the TEC data treatment? Are there some new assessments of the models taking in account the plasmosphere impact? All these subjects were examined by many authors earlier (as the authors themselves reviewed in the Introduction section). Thus I would recommend to the authors to rewrite abstract and introduction section in such a manner to explain the above mentioned items clearer.
  • Section 2, Lines 159-160: “To eliminate the errors from multipath, we also remove the satellite and receiver biases from the VTEC values”. First, this sentence contains a repetition of the text in lines 132-136. Second, please explain how it is possible to remove the multipath errors from the VTEC time series by means of DCB removing? Do these processes really relate one to another?
  • Section 2, Lines 162-165: According to this text the authors used “…. the monthly mean values of the diurnal GPS-TECs.” I do not understand how is it possible to provide comparative analysis of the ionospheric models based on the monthly mean values of the diurnal TEC?
  • Page4, Lines 178-179: “..by averaging the diurnal monthly mean values..”. I do not understand what “diurnal monthly mean” means. Please rebuild the text of the second section and explain the procedure of the TEC data treatment clearer.
  • Section 3: There is no necessary to split the section up with tiny sub-sections. Most of the subsections do not contain adequate explanations of the obtained results, but only fact statements. I would recommend to the authors: 1) join all these tiny sub-sections in entire section text; 2) add clear comparative analysis between the considered cases.

Minor comments

  • Page 1, Line4: Please remove dot at the end of headline
  • Page 1, Line19: Please remove the brackets before and after GUAM and rewrite the sentence in correct English
  • Page 1, Lines 19-21: The sentence sound very unclear. I do not understand whether the authors mean “…shape of the electron density profile….” within the equatorial anomaly region or outside one? Please rewrite the sentence in clearer manner.
  • Page 1, Line 23: What do the authors mean as “The deviation of NeQuick model”
  • Lines 31-32: “… trans-ionospheric communication systems such as satellites, aircraft, and surface transportation…”. This sentence sounds very strange. Please rewrite it.
  • Page 2, Line 46: please replace “Models” with “models”.
  • Page 2, Line 59: Please remove “66”
  • Page 2, Line 59: “…to the orbital height of the satellite…”. Which satellites do you mean? LEO, MEO or GEO?
  • Page 2, Line 60: “…and Technical research (COST) 238 regional electron content model…”. What do you mean here? Please rewrite it in correct English.
  • Page 2, Line 78: “..IRI-Corr…”. What the model do the authors mean? It there official name of this model and reference on it?
  • Page 2, Line 90: “…are fairly accurate in trends with GNSS measurements…”. What do the authors mean? Please rewrite the sentence in correct English.
  • Page 2, Line 91: “…performed better than IRI-Plas except in DAV1, KERG, and ADIS.” What do the authors mean? Please rewrite the sentence in correct English.
  • Page 3, Line 102 (and the same in sense in 115 and 116): “…eight GPS-TEC stations in the Northern and southern hemisphere over the Globe..”. Earlier the authors announced (see the Abstract) that they are going to examine “…8 GPS stations within the equatorial and low latitude…”. What it the subject of the author’s research?
  • Page3, Line 103: Please replace “our” with “Our”.
  • Page3, Line 114: “…NeQuick-2 model 123 …”. What do the authors mean? Please rewrite the sentence in correct English.
  • Page3, Line 119: “…2013 and 2014 128…”. What do the authors mean? Please rewrite the sentence in correct English.
  • Page 3, Line 128: “…world indicating the location of GPS receivers globally…”. What do the authors mean? Please rewrite the sentence in correct English.
  • Page 3, Lines 134-135: “…removes satellite and receiver biases from Differential Code Bias (DCB) IGS code files…”. What do the authors mean? Please rewrite the sentence in correct English.
  • Pages 3-4, Lines 136-157: This is waste information. I would recommend to the authors to replace this text with common and well known VTEC and STEC equations and correspondent references.
  • Page 4, Lines 166-167: “…based on all available data sources used for their development.” What do the authors mean? Please rewrite the sentence in correct English.
  • Page 4, Lines 166-172: This is absolutely unclear text. Please rebuild it and rewrite it in correct English
  • Page 7, Lines 326-327: “…2013 and 2014, a year of increase in solar activity…”. What do the authors mean? Please rewrite the sentence in correct English.

Round 2

Reviewer 1 Report

Corrections are made in accordance with the comments made by me. Thanks.

Author Response

Thanks for the Referee who has no further comments.

Reviewer 3 Report

Dear Authors,

Please explain what do you mean as "low latitudes" as the area of your research. I suppose that the Antarctica (DAV1) station you considered is set in the polar region but not in the "low latitudes". Please add you detaled comments about this item in the second section of your manuscript.
